# Usefulness and Validity of a Jaw-Closing Force Meter in Older Adults

**DOI:** 10.3390/geriatrics7060145

**Published:** 2022-12-19

**Authors:** Mina Kawashima, Kanako Yoshimi, Kazuharu Nakagawa, Kohei Yamaguchi, Miki Ishii, Shohei Hasegawa, Rieko Moritoyo, Ayako Nakane, Haruka Tohara

**Affiliations:** Department of Dysphagia Rehabilitation, Division of Gerontology and Gerodontology, Graduate School of Medical and Dental Sciences, Tokyo Medical and Dental University, Tokyo 113-8510, Japan

**Keywords:** jaw-closing force, occlusal force, jaw-closing force meter, older adults, oral function

## Abstract

We prototyped a new device with a soft and flexible pressure sensor to measure the force to close the mandible with or without occlusal support (jaw-closing force, JCF). This study aimed to clarify the practicality of this instrument. Healthy young and older adults with occlusal support were recruited. Intra- and inter-rater reliability of the JCF meter was examined using data from younger participants. Data regarding age, sex, body mass index, remaining teeth, and dentures of the older adults were obtained. Furthermore, the right and left JCFs were measured using a JCF meter; occlusal force was measured using an existing occlusal force-measuring device. Intra- and inter-rater correlation coefficients were significantly reproducible (0.691–0.811, *p* < 0.05). JCF was correlated with occlusal force (*p* < 0.05). Multiple regression analysis revealed that factors significantly associated with JCF included denture status (*p* < 0.001), age (*p* = 0.038), and occlusal force (*p* = 0.043). The prototyped JCF meter can measure JCF with high reproducibility, reliability, and validity. Further, association with occlusal force, which is an existing index, was observed. This device could be used to measure the JCF with or without occlusal support as a new method of evaluating oral function in older adults.

## 1. Introduction

The global population is aging rapidly. The effects of aging on the body are diverse, and oral function is one of the physical functions that decline with age [1]. As older adults lose teeth, their occlusal force declines [2]. It has been reported that occlusal force is a predictor of cognitive decline [3] and is also related to motor function and falls [4]. Occlusal force also declines with decreased occlusal support, and denture use [5,6], which limits the form and firmness of foods that can be consumed [7]. Clinically, however, there are cases where dentures should be worn but are not due to cognitive decline, oral dysfunction, or impaired hand motor function [8]. Notably, it has been reported that occlusal force is significantly related to food and nutritional intake in older adults rather than the number of teeth, and is an indicator of masticatory function [9]. In order to consider the recommended food texture for such geriatric patients, we thought it would be clinically significant to measure the force required to close the mandible in older people who have lost occlusal support (hereafter referred to as “jaw-closing force”).

Occlusal force is the pressure that occurs when the upper and lower teeth bite into each other. Currently used instruments for evaluating occlusal force include a device that visualizes occlusal pressure and occlusal balance by biting a pressure-sensitive sheet over the entire dentition and reading it with a dedicated scanner [10]. Another device implemented quantifies the occlusal force when biting the tip of the device with the molar part on one side [11]. Both these instruments are applicable only to those who have occlusal support, and it is impossible to measure occlusal force in edentulous people or those who lack occlusal support due to missing teeth.

Based on our clinical experience, there are a significant number of people with edentulous jaws who eat without dentures. We aimed to assess the difference between people who are functionally able to consume a normal diet and those who are not in the absence of posterior teeth and no dentures. Although it is clear that the tongue compensates for masticatory movements in older adults with missing teeth [12], it is unclear whether the edentulous ridge compensates for tooth function. When evaluating mastication-like movement in people without teeth, we hypothesized that an indicator other than tongue function, namely the ability to crush food between the jaw crests or jaw-closing force, would be useful.

Therefore, we requested a device that can measure the jaw-closing force by biting between the edentulous ridge even in those without occlusal support, and a prototype was manufactured by Murata Manufacturing Co. Ltd. Our aims were to clarify the reliability and validity of the prototype jaw-closing force meter (JCF meter) in healthy participants with occlusal support and to determine the correlation between the jaw-closing force and the existing index of occlusal force.

## 2. Materials and Methods

### 2.1. Participants

This was a prospective cross-sectional study. We included 13 healthy young volunteers (5 males and 8 females, 32.0 ± 6.6 years), and 62 healthy older adults (38 males and 24 females, median (interquartile range, (IQR); 73.5 (70.3, 80.0)) who participated in health surveys conducted in December 2021 in Yokohama, May 2022 in Tokyo, and December 2022 in Kyoto. The participants were recruited one month before each survey. Healthy participants were defined as those able to sign informed consent and with normal self-reported swallowing ability. The exclusion criteria were as follows: patients who did not consent to participate in the study; a history of temporomandibular disorder, cognitive disorder, neurologic disease, or head and neck cancer or surgery; and tooth or gingival pain in the posterior teeth. The study protocol was approved by the Dental Research Ethics Committee of Tokyo Medical and Dental University (Approval ID: D2020-024). All participants provided informed consent prior to enrollment, and written informed consent was obtained for participation in this study. This study was conducted ethically in accordance with the World Medical Association Declaration of Helsinki.

### 2.2. The Jaw-Closing Force Meter (JCF Meter)

This is a prototype developed by Murata Manufacturing Co., Ltd. (Kyoto, Japan). A flexible pressure sensor measuring 30 mm long, 15 mm wide, and 8 mm thick is attached to the tip of the body of the instrument (Figure 1A). When used, a special protective film is attached to the flexible pressure sensor. The approximate shape of the protective film is that of a trapezoid with a top side of 25 mm, a bottom side of 60 mm, and a length of 85 mm, and is made of polyethylene film processed into a bag shape. A sponge made of porous olefin material, 30 mm long, 24 mm short, and 3 mm thick, is attached to the upper and lower sides of the film (Figure 1B). When pressure is applied to the sensor, the display in the center of the unit shows the pressure in N (Figure 1C).

### 2.3. Data Collection

#### 2.3.1. Reliability of the JCF Meter

The jaw-closing force was measured by two dentists on healthy young volunteers. The measurements were performed with the participants seated in a chair. First, the central part of the flexible pressure sensor of the JCF meter was placed on the first molar of the participant’s lower jaw. The participant was then instructed to bite with maximum force. The maximum jaw-closing force was recorded as the value at the point when the value shown on the display stopped increasing and stabilized. Measurements were obtained three times on each side. One of the two dentists also measured the same participant’s jaw-closing force using the JCF meter after a 1-week interval.

#### 2.3.2. Examination of Factors Correlated and Related to Jaw-Closing Force

Healthy older participants with occlusal support were interviewed about their age and sex. Height and weight were measured, and body mass index (BMI) was calculated. Remaining teeth and the use of dentures in the upper and lower molars were evaluated. Occlusal status with dentures and prosthetics was evaluated using the Eichner classification [13]. The occlusal support regions were classified into four areas: the right and left premolars, and the right and left molars. The participants were classified into three groups: Eichner A (occlusal support in all four areas), Eichner B (occlusal support in one, two, or three areas), and Eichner C (no occlusal support).

Occlusal force was measured using a Dental PrescaleII^®^ (GC Co., Ltd., Tokyo, Japan). With the participant seated in a chair, the pressure-sensitive film was placed in maximum occlusion, and the occlusal forces on the right and left sides were calculated using a dedicated software. As for the jaw-closing force, the participant was instructed to bite with a maximum extent three times on each side using the JCF meter while sitting in a chair. The median value was calculated from the three measurements for each side.

### 2.4. Data Analysis

To examine the reliability and validity of the JCF meter, the intra- and inter-rater correlation coefficients were calculated from the data of healthy young volunteers.

Subsequently, we examined the associations between the jaw-closing force and each of the items. The correlation between age, sex, BMI, occlusal force measured with a dental prescale, and jaw-closing force (edentulous jaw, no molar occlusion) was examined using Spearman’s rank correlation coefficient. To further examine the factors that correlate with closing force, multiple regression analysis was conducted with the objective variable as the jaw-closing force on the right side and five explanatory variables: age, sex, BMI, occlusal force (dental prescale), and denture status. The sample size was 58 subjects when power was 0.8, α = 0.05, effect size 0.25, and the number of explanatory variables was 5 [14]. The significance level was set at 5% or less. Statistical Package for Social Sciences (SPSS) version 25 (IBM Japan, Tokyo, Japan) was used for statistical analysis.

## 3. Results

### 3.1. Reliability and Validity of the JCF Meter

Intra-rater (n = 13) and inter-rater (n = 13) reliability are shown in Table 1. All participants were able to be measured without any trouble. The mean age of the healthy young participants was 32.0 ± 6.6 years, and there were five males. The intraclass correlation coefficients (ICC) were calculated: intra-rater reliability was 0.81 (*p* = 0.003) for the right measurement and 0.691 (*p* = 0.023) for the left measurement; inter-rater reliability was 0.706 (*p* = 0.007) for the right measurement and 0.789 (*p* = 0.006) for the left measurement. Both intra- and inter-rater reproducibility were high.

### 3.2. Correlation of Each Item with the Ability to Close One’s Jaw

The characteristics of the healthy older adults are shown in Table 2. All participants were able to be measured without any trouble. No one was unable to be measured due to pain or had to stop due to pain during the measurement. The median (IQR) for age was 73.5 (70.3, 80.0). The right side of both occlusal force and jaw-closing force showed slightly higher values than the left side.

Table 3 shows the correlations between the jaw-closing force and each item in healthy older participants. Both occlusal force and jaw-closing force were moderately correlated between the right and left sides (OF: r = 0.684, *p* < 0.001; JCF: r = 0.712, *p* < 0.001). The correlation r ranged from 0.298 to 0.433 for both the left and right sides. Only the jaw-closing force on the right side correlated with age (r = −0.413, *p* < 0.001). However, the jaw-closing force did not correlate with BMI.

### 3.3. Examination of Factors Associated with Jaw-Closing Force

After adjustment for confounders, denture status (β = −0.43, *p* < 0.001), occlusal force (β = 0.22, *p* = 0.043), and age (β = −0.24, *p* = 0.038) were associated with the jaw-closing force. Sex and BMI were not associated with the jaw-closing force (Table 4).

## 4. Discussion

### 4.1. Reliability of the Prototyped JCF Meter

Intra-rater reliability showed a difference in the number of intra-class correlations between the left and right sides of the participant. When the examiner holds the JCF meter in the right hand and measures the subject’s right-side jaw-closing force, it is easy to insert the tip of the JCF meter into the rear of the mouth because the left hand can retract the lips. However, when measuring the jaw-closing force on the left side, the participant may have switched the JCF meter to the left hand to eliminate the lips with the right hand, or may have measured the jaw-closing force with the lips not fully eliminated. The correlation coefficient was 0.811 for the right side and 0.691 for the left side for the same examiner, with the left side having a lower correlation coefficient, which may be related to the operability of the instrument.

The measurers were right-handed, and during the JCF measurement on the left side, they stood on the participant’s left side and operated the equipment with their non-dominant left hand. As a result, if the sensor was improperly positioned to bite or if the instrument was not held stably during the measurement, there is a possibility that the measurement results may vary. To improve this situation, it may be useful to mark the biting position of the sensor on the protective film in advance so that the bite force would be applied at the same position and angle, and the sensor could be designed to be L-shaped so that the device could be inserted into the mouth from the front of the participant. However, when considering the reliability of continuous data, an ICC of 0.6 or higher is required, and 0.8 or higher is more desirable [15]. Therefore, based on the results of this study, the newly developed JCF meter is sufficiently reliable. The inter-rater reliability showed smaller differences between the left and right sides than the intra-rater reliability. The correlation coefficients for both left and right sides were more than 0.7, indicating that stable measurement is possible even if the examiner changed.

### 4.2. Characteristics of Prototyped JCF Meter and Comparison with Existing Occlusion Meters

In this study, the JCF meter was used for participants with occlusal support. The conventional occlusal force meter GM10^®^ (Nagano Keiki Co., Ltd., Tokyo, Japan) is similar in shape to the JCF meter; however, the sensor at the tip to measure occlusal force and the protective film are both made of a hard material. On the other hand, the JCF meter is a combination of a flexible pressure sensor and a protective sponge, which is made of soft material and changes shape when pressure is applied by the teeth or edentulous ridge. To date, no JCF meter developed uses a flexible pressure sensor. Therefore, it was necessary to first evaluate the reliability of the flexible pressure sensor itself to examine whether an accurate measurement was possible.

The jaw-closing force was smaller than the occlusal force. The occlusal force is the sum of the pressures on the occlusal contacts from the anterior to the posterior teeth. Meanwhile, the JCF meter receives forces at the occlusal surfaces of the molars or at the jaw crest. Therefore, the force that occurs at maximum occlusion may have been the same value but could have caused a difference as a measured value.

The original purpose of the JCF meter is to measure the bite force between the edentulous ridge or between the edentulous ridge and teeth without occlusal support. Therefore, it is necessary to investigate the effectiveness of the flexible pressure sensor for those without occlusal support. In cases such as when occlusal support is lost or edentulous individuals have a gap between the teeth and the edentulous ridge or between the upper and lower edentulous ridge, even when the mouth is closed to the maximum extent, it is impossible to occlude a thin pressure-sensitive sheet such as the Dental PrescaleII simultaneously on both sides. Therefore, the JCF meter with a thick flexible pressure sensor and a protective film can measure the force to close the jaw on each side by crushing it with the teeth or the edentulous ridge. For more reliable measurement, it is desirable to standardize the insertion position of the sensor into the mouth, the position of the sensor in occlusion, and to place the sensor such that it is well clear of the lips and perpendicular to the edentulous ridge or dentition.

### 4.3. Factors Associated with Jaw-Closing Force

Jaw-closing force as measured by the JCF meter negatively correlated with age. This is thought to be due to the decline in strength of the perioral muscles with age. A positive correlation was also observed between the jaw-closing force and occlusal force, which was significantly related. When closing the mouth, the masseter muscle, which is the jaw-closing muscle, raises the mandible. Since occlusal force is strongly related to the masseter muscle mass [16], jaw-closing force can be viewed as an indirect assessment of the masseter muscle mass.

The JCF meter has a flexible pressure sensor thickness of 8 mm and a protective film thickness of 6 mm, for a total thickness of 14 mm. This thickness is assumed to compensate for the gap between the edentulous ridge or between the edentulous ridge and the teeth and to be able to measure the force to close the mouth in patients with edentulous jaws. In this study, the jaw-closing force and occlusal force were correlated, and multiple regression analysis revealed that they were related, indicating that the JCF meter is effective as a new oral function measurement device. Moreover, it is known that the use of dentures decreases occlusal force [6], and the same was true for jaw-closing force, which was strongly related to the presence or absence of dentures.

Based on these results, it is possible to evaluate the values of jaw-closing force and occlusal force of older edentulous patients as similar; although, the measured values (N) do not coincide. With regard to the muscle mass and strength of the masseter muscles, an association with grip strength [17], an association with increased risk of physical dysfunction independent of confounding factors such as the number of remaining teeth [18], and an association between occlusal force and vascular endothelial function [19] have been reported in older persons living in the community. The relationship between oral function and not only perioral muscles but also whole-body muscular strength and function has been elucidated, and evaluation of jaw-closing force with the JCF meter may be useful for health management of older adults.

### 4.4. Limitations

In this study, we compared the usefulness of the prototyped JCF meter with existing occlusal force-measuring instruments. Since the participants were either those with occlusal support or those using dentures, jaw-closing forces between edentulous ridges and teeth or edentulous ridges were not measured. Therefore, we could not establish a value that can be used as an indicator of the maintenance or decline of oral function. We plan to examine the usefulness of the JCF meter for those who do not have occlusal support. Further, by examining the relationship between jaw-closing force and masticatory ability and recommended food forms, it will be possible to establish values that can be used as an indicator of the ability of persons without occlusal support to crush food and as an indicator of the maintenance or decline of oral function.

In addition, further study is needed on the structure of the equipment and performance evaluation. With the shape of the sensor of the prototype JCF meter, the number of cases that can be measured is considered to be limited at present. The total thickness of the soft sensor and the protective film is 14 mm. Therefore, if the gap between the upper and lower edentulous crests is large, such as when the edentulous crest is highly resorbed, the sensor part may not be thick enough to apply adequate force and may not be able to measure properly. Moreover, if there is tooth protrusion or if the amount of mouth opening cannot be obtained sufficiently, the sensor cannot be inserted into a certain position, and measurement may be difficult. As for the flexible soft sensor, it is necessary to verify whether the value changes depending on the bite position and whether different thicknesses of the protective film affect the value. In the future, we plan to consider combinations of flexible pressure sensors and protective film thicknesses to improve the versatility of the JCF meter in order to accommodate people with various tooth loss and edentulous crest morphologies.

## 5. Conclusions

The prototyped JCF meter is capable of measuring jaw-closing force with high reproducibility, reliability, and validity. The jaw-closing force, which has a value that correlates with the existing index of occlusal force, could be used with or without occlusal support, and might be beneficial as a new index of oral function in older adults.

## Figures and Tables

**Figure 1 geriatrics-07-00145-f001:**
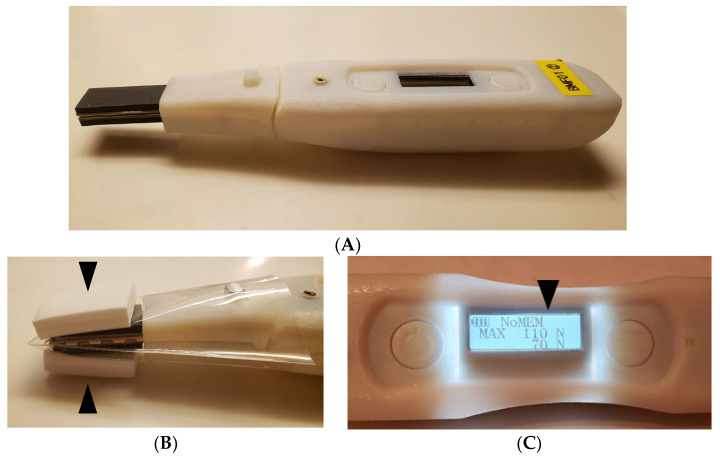
Jaw-closing force meter: (**A**) The arrowhead indicates a flexible pressure sensor; (**B**) protective film is attached. The arrowheads indicate porous material attached outside of the protective film; and (**C**) the arrowhead indicates the closing force on the display.

**Table 1 geriatrics-07-00145-t001:** Intra- and inter-rater reliability of the force-closure instrument (n = 13).

		Intraclass Correlation Coefficient	95% Confidence Interval	*p*-Value
Intra-rater reliability	Right	0.811	[0.406, 0.942]	0.003
	Left	0.691	[0.027, 0.905]	0.023
Inter-rater reliability	Right	0.706	[0.105, 0.905]	0.007
	Left	0.789	[0.309, 0.932]	0.006

**Table 2 geriatrics-07-00145-t002:** Characteristics of participants (n = 62).

Characteristics		
Age (years), median (IQR)		73.5 (70.3, 80.0)
Sex, N (%)	Male	38 (61.3)
	Female	24 (38.7)
Eichner classification, N (%)	A	59 (95.2)
	B	2 (3.2)
	C	1 (1.6)
BMI (kg/m^2^), mean ± SD		22.5 ± 2.7
OF: R (N), median (IQR)		352.6 (209.6, 458.5)
OF: L (N), median (IQR)		300.1 (196.5, 451.1)
JCF: R (N), median (IQR)		326.7 (210.0, 466.7)
JCF: L (N), median (IQR)		280.0 (158.3, 422.5)

BMI, body mass index; IQR, interquartile range; OF: occlusal force, JCF: jaw-closing force, R, right; L, left.

**Table 3 geriatrics-07-00145-t003:** Correlation coefficients for each item (n = 62).

	Age	BMI	OF (R)	OF (L)	JCF (R)	JCF (L)
**Age**	1					
**BMI**	0.119	1				
**OF (R)**	−0.312 *	−0.042	1			
**OF (L)**	−0.165	−0.045	0.684 **	1		
**JCF (R)**	−0.413 **	−0.103	0.433 **	0.299 *	1	
**JCF (L)**	−0.190	−0.113	0.303 *	0.298 *	0.712 **	1

BMI, body mass index; OF, occlusal force; JCF, jaw-closing force; R, right; L, left. Spearman rank correlation coefficient, * *p* < 0.05, ** *p* < 0.001.

**Table 4 geriatrics-07-00145-t004:** Multiple regression analysis with jaw-closing force as dependent variables (n = 62).

	Standardizing Coefficient (β)	*p*-Value	VIF
**Age**	−0.24	0.038*	1.23
**Sex**	−0.19	0.070	1.05
**BMI**	−0.10	0.344	1.02
**OF**	0.22	0.043 *	1.11
**Presence of denture**	−0.43	<0.001 **	1.25

BMI, body mass index; OF, occlusal force; VIF, variance inflation factor, Multiple regression analysis, Adjusted R^2^, 0.389; * *p* < 0.05, ** *p* < 0.001.

## Data Availability

The data that support the findings of this study are available on request from the corresponding author, K.N. The data are not publicly available due to their containing information that could compromise the privacy of research participants.

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
