# Peer review of "Usefulness and Validity of a Jaw-Closing Force Meter in Older Adults"

_geriatrics, 2022, doi:10.3390/geriatrics7060145_

Round 1

Reviewer 1 Report

Dear Authors

The purpose of the study is to test and evaluate the efficiency of a prototype masticatory force meter in healthy subjects. Therefore, the work fits into an unexplored field of literature and provides the possibility of knowing a new device that is useful both from an academic and clinical point of view Certainly a limitation of the work, especially considering that it is a longitudinal study, is to have a limited and not well selected sample. there is a different number of children and adults. It is not clear how the concept of healthy subjects is defined and the time of recruitment is not specified. The tables are well presented and help comprehensively to understand the research results. A better job could be done with the images: they are not of high quality and are not conducive to understanding the device The conclusions clearly and exhaustively report what emerged from the results, highlighting the effectiveness and usefulness of the prototype examined in the work.

the manuscript is interesting and suitable for the newspaper although some lacks in method and design have emerged

INTRODUCTION

Please provide an explicit statement of the objectives or questions the review addresses.

The MATERIALS AND METHODS section presents some lacks:

- Please specify if approval of the Ethical Committee is present and add it to the material and methods section

- Please indicate if the power analysis has been made

-  A sample size calculation is necessary for a longitudinal cross-sectional study, please add

RESULTS

this section is well-supported and aided by tables

DISCUSSION

The discussion is too little developed, it should be implemented with some considerations and reflections on what emerged from the analyzed data

Reviewer 2 Report

This manuscript reports an interesting study and shows us meaningful findings about Jaw closing force that are important for eating. The topic addressed is interesting and deserves a constructive discussion. The paper is well-written, the tables and figures are of high quality, and the authors have clearly worked hard to produce a comprehensive dataset and detailed description of their methods.

Abstract

The Abstract appropriately summarizes the manuscript. There are no discrepancies between the Abstract and the manuscript. The Abstract is easy to understand without reading the manuscript.

Introduction

The length of Introduction section is appropriate and well-written about past papers.

Method

L62-70

I think that the description of the subject's tooth condition should be added.

And I think that exclusion criteria should be added.

L75-76 The shape of protective film is hard to understand. “60mm long…and 85 mm long” which one is true? Is that trapezoid?

Results

Were all participants able to measured without trouble? If there are people who could not be measured, it should be noted.

Discussion

If, as you point out, usability makes the left side unreliable, do you have any ideas on how to improve it?

About the whole paper

Mechanical considerations, such as the effective range of the film sensor, should be done separately from this research. It is also necessary to consider whether the thickness of this sensor can accommodate the intermaxillary distance of edentulous patients.
